# Splitting Tensile Test of ECC Functional Gradient Concrete with PVA Fiber Admixture

Yin Xu [1], Qiang Liu [2], Xiaoqiang Zhang [3], Xiaofeng Xu [4] and Peng Liu [5,6,*]

1. Wuhan City College, Huangjia Dawan, Wuhan 430075, China; yinxucs@sina.cn
2. Wuhan United Investment Real Estate Co., Ltd., 399 Wenhua Avenue, Wuhan 430212, China; liuqiang1@whltzy.com
3. Jinan Municipal Engineering Design & Research Institute (Group) Co., Ltd., Jinan 250013, China; zhszxq@jnszy.com
4. Hubei United Investment Group, 99 Zhongnan Road, Wuhan 430000, China; xuxiaofeng007008@126.com
5. School of Civil Engineering, Central South University, 22 Shaoshan Road, Changsha 410075, China
6. National Engineering Research Center for High-Speed Railway Construction, Changsha 410075, China
* Correspondence: liupeng868@csu.edu.cn

**Abstract:** Engineered cementitious composite (ECC) functional gradient concrete has a promising application future, and its mechanical features are piquing the interest of researchers. The impacts of this strength class of concrete, interface reinforcement technique, ECC thickness (i.e., fiber dosage), and other factors on the splitting tensile strength qualities are explored using an experimental investigation of functional gradient concrete. The splitting tensile tests of 150 mm × 150 mm × 150 mm functional gradient concrete specimens were used to explore the link between concrete strength grade, interface reinforcing technique, and ECC thickness with polyvinyl alcohol (PVA) fiber additive and functional gradient concrete. The test results show that the splitting tensile strength of functional gradient concrete increases as the concrete strength grade increases; different interfacial treatments have a significant effect on the splitting tensile strength of functional gradient concrete; and the effect of ECC thickness change on the splitting tensile strength of functional gradient concrete shows different trends, which research can be used as an experimental reference for functional gradient concrete engineering applications.

**Keywords:** functional gradient concrete; engineered cementitious composites (ECC); polyvinyl alcohol (PVA) fiber incorporation; splitting tensile properties; concrete strength; interfacial reinforcement





## 1. Introduction

Concrete materials are commonly utilized in civil engineering, roads, bridges, and other engineering domains. However, it is important to acknowledge the presence of defects in concrete that can hinder its application. For instance, the tensile strength of steel reinforcement combined with concrete is significantly lower (only 1/8–1/12 of the compressive strength), leading to early corrosion of the longitudinal reinforcement. Additionally, the inherent brittleness of concrete further exacerbates these defects, impeding the progress of concrete construction materials. Currently, engineered cementitious composites (ECC) replace traditional ordinary concrete and are applied in engineering structures, one of the main ways to solve the above problems [1,2]. The reason lies in the thickness of the ECC protective layer area, which is made of ECC material instead of the original concrete. By harnessing the nonlinear deformation, energy absorption, and crack control properties of the ECC and optimizing the interface between ECC and concrete to enhance its bond performance, it transforms into a functional composite material [3], which results in enhanced load-carrying capacity and ductility of the structural element, as well as improved control over crack width, thereby extending the service life of the element [4,5].

On the other hand, ECC, which stands for strain-hardening cementitious composites and pseudo-strain cementitious composites, refers to a distinct kind of concrete that has

both high tensile strength and ductility. The material has a tensile strain capacity above 3% while retaining a fiber volume fraction of no more than 2% [6]. It is well known that determining the axial tensile strength of plain concrete by a direct axial tensile test is necessary and also difficult. The reason is that there are inevitable problems, such as skewing and eccentricity during the installation of plain concrete specimens, and their geometric and physical centers do not coincide with each other. Thus, the test results determined by the direct axial tensile test fluctuate greatly. Furthermore, researchers have conducted a range of empirical and theoretical investigations on the tensile characteristics and ontological connections of ECC [7–9]; however, a cohesive consensus has not yet been established. Given this, in order to obtain the axial tensile strength of ECC, the split tensile strength test was considered to reflect the axial tensile strength of ECC indirectly. The test results obtained from this splitting tensile strength test are less discrete and simple to implement.

Research on the splitting tensile strength of functionally graded concrete has shown that several parameters influence its splitting tensile strength. There are usually factors such as the strength class of concrete [8], interfacial reinforcement process [9], the thickness of ECC [10], polyvinyl alcohol (PVA) fiber admixture [11], and age [12]. It was found that the splitting tensile strength of functionally graded concrete increased as the strength grade of the concrete increased [13]. Tian et al. [10] concluded that the roughness of old concrete, the strength of the old concrete matrix, the type of interfacial agent and binder, and the age have different degrees of influence on the bond tensile strength through the experimental study of bonding tensile properties of ultra-high toughness cementitious composite with concrete.

The damage of functional gradient concrete mostly occurs at the interface bond, and the interface becomes a key factor affecting the splitting and tensile properties of functional gradient concrete. Ibrahim et al. [14] pointed out that the main influencing factors of the bonding properties of new and old concrete are the interface treatment method and state, the type of interfacial agent, the bond strength of reinforced concrete, and the difference in deformation of new and old concrete. Yin and Liew [15] concluded that interfacial properties significantly affect both mechanical properties and damage modes of composites, and interface design has always been an important part of fiber-reinforced composites microstructure design research. Qian et al. [16] proposed that there exists a transition layer between the old and new concrete interfaces consisting of three thin layers: a penetration layer, a strong effector layer, and a weak effector layer, where the surface condition of the strong effector layer, the interfacial agent, and the repair material together determine its performance. Jiang et al. [17], through the study of shear properties of steel fiber cement mortar bonded to concrete, concluded that the type of interfacial treatment, the strength of steel fiber cement mortar, and the strength of the old concrete can significantly increase the bond surface shear strength. A study by Njim et al. [18] found that the use of an artificial notch treatment significantly affected the splitting tensile strength of functionally graded concrete. He et al. [19] tested the mechanical properties of old and new concrete binders. The effects of interfacial roughness and interfacial binder on the bonding properties of old and new concrete were investigated. The study showed that the transition zone between old and new concrete is the key to bonding old and new concrete. The interfacial roughness affects the penetration layer, the interfacial binder improves the reaction layer, and the proper improvement of the transition zone between old and new concrete is conducive to better bonding of old and new concrete. In their experiments on the bond surface of new and old concrete, Zhang et al. [20] and Manawadu et al. [21] examined the impact of bond surface roughness, age, and interfacial agent on fracture toughness. The analysis revealed that the fracture toughness of the bond surface increased substantially as the roughness of the surface increased.

Additionally, the fracture toughness of the bond surface increased in a hyperbolic fashion as the age of the bond increased. The fracture toughness of both new and old concrete increased in a hyperbolic fashion as the bonding age increased. Furthermore, the

inclusion of an interfacial agent had a substantial impact on enhancing the bond fracture toughness of both new and old concrete. Zhang et al. [22] suggested that the bonding performance between old and new cement mortar mainly depends on the following factors: first, the void area of the old cement mortar stone surface; second, the number and area of the cement particles in the new cement mortar in contact with the old mortar stone surface; and third, the microstructure of the new cement mortar formed at the bonding interface. Then, the following methods are also proposed to improve the bonding strength: first, the old mortar stone surface is soaked in water to facilitate the hydration of the new cement mortar; second, the use of fine particles, grading of cement or other materials as an interfacial agent, the choice of interfacial agent requires its own good performance, dense structure, and can be generated after the hydration of the crystals that can improve the adhesive force; third, pay attention to the quality of the construction, to ensure that the new and old materials indicate that the new and old materials are in close contact.

A study by Feng et al. [23] found that the roughness of old concrete surfaces and the strength of encapsulated steel fiber cement mortar significantly affected the interfacial bond strength of new and old materials. Steel fibers mixed in the repair material can be a certain degree of the bond strength of the new and old materials.

The existing experimental studies on the splitting and tensile properties of functional gradient concrete have achieved certain results, but some aspects still need improvement. Regarding the sample preparation methods, some studies focused on the splitting tensile properties of functional gradient concrete. Still, the current sample preparation methods may not accurately simulate the functional gradient concrete structures in real engineering [24,25]. Therefore, improvement in sample preparation methods is needed to reflect the real performance of functional gradient concrete.

Regarding the test setup and loading method, the current test setup and loading method may not be able to adequately consider the non-uniformity and gradient of functional gradient concrete [26,27]. Therefore, there is a need to design a more appropriate test setup that can simulate the stresses in real projects and consider the special characteristics of functional gradient concrete.

Regarding the selection of test parameters, the performance of functional gradient concrete is affected by various factors, including material composition, gradient distribution form, and gradient change rate [28,29]. The current study needs to clarify further and optimize the selection of test parameters to obtain more accurate data on splitting tensile properties.

Regarding the analytical methods, current experimental studies on the splitting and tensile properties of functionally graded concrete mainly focus on determining mechanical property data, and there are still fewer analyses of the macroscopic and microstructures [30,31]. Therefore, there is a need to develop more comprehensive methods for analyzing the results to gain a deeper understanding of the performance and damage mechanisms of functional gradient concrete.

Regarding sustainability and durability, the study of sustainability and durability of functional gradient concrete as a new material is also very important [32–35]. The material's long-term performance, durability, and environmental adaptability should be considered in the experimental study of splitting and tensile properties to assess its feasibility and application prospects in practical engineering.

The specimen size in this study is 150 mm $\times$ 150 mm $\times$ 150 mm, the thickness of the ECC material is 75 mm and 45 mm, and the thickness of the matching regular concrete is 75 mm and 105 mm. The effect of three parameters on concrete's splitting and tensile characteristics, namely concrete strength grade, interfacial reinforcement, and ECC thickness, is examined and analyzed, as well as the trend of the effect on the split tensile properties. The influence of the three parameters on the concrete's splitting tensile characteristics was investigated and analyzed. To investigate and analyze the concrete strength grade, interface enhancement technology, the thickness of ECC, and other aspects of the split tensile strength performance using the split tensile strength test. At the same

time, the splitting tensile strength under the condition of numerous influencing elements was examined in this article, which gives the experimental reference value for its application in the engineering sector.

## 2. Experimental Design

### 2.1. Testing of Raw Materials and Related Mixing Ratios

The ECC material used in this study is prepared by ordinary silicate cement P.O 42.5, domestic water, class I fly ash, quartz sand with a particle size of 100–200 mm, a poly-carboxylic acid-based high-efficiency water-reducing agent with a water-reducing rate of 20%–40%, and domestically-produced PVA fibers, and its mixing ratios are shown in Table 1. The performance indexes of domestically produced PVA fibers are shown in Table 2. Some of the tested raw materials are shown in Figure 1. Fly ash is composed of $SiO_2$, $Al_2O_3$, $Fe_2O_3$, and CaO. For further details, please refer to the website http://www.glhuayue.com/index.php?aid=469 (accessed on 27 January 2024). The chemical makeup of silica fume consists mostly of amorphous $SiO_2$, which makes up 92.4% of its composition. For further details, go to the website http://www.aerbadi-nxyl.com/custom/a/33 (accessed on 27 January 2024). Notably, a fine steel wire mesh is utilized in this experiment at a spacing of 20 mm.

**Table 1.** Performance index of domestic PVA fiber.

| Length (mm) | Diameter (μm) | Modulus of Elasticity (GPa) | Elongation % | Tensile Strength (MPa) | Density (g/cm$^3$) |
|---|---|---|---|---|---|
| 12.0 | 12–18 | 35.0 | 6–8 | 1200.0 | 1.3 |

**Table 2.** Ratio of ECC materials (unit: kg/m$^3$).

| Cement | Fly Ash | Silica Fume | Quartz Sand | Water | Water Reducer | Fiber (Volume Ratio) |
|---|---|---|---|---|---|---|
| 240.0 | 720.0 | 240.0 | 432.0 | 420.0 | 4.8 | 26.0 |

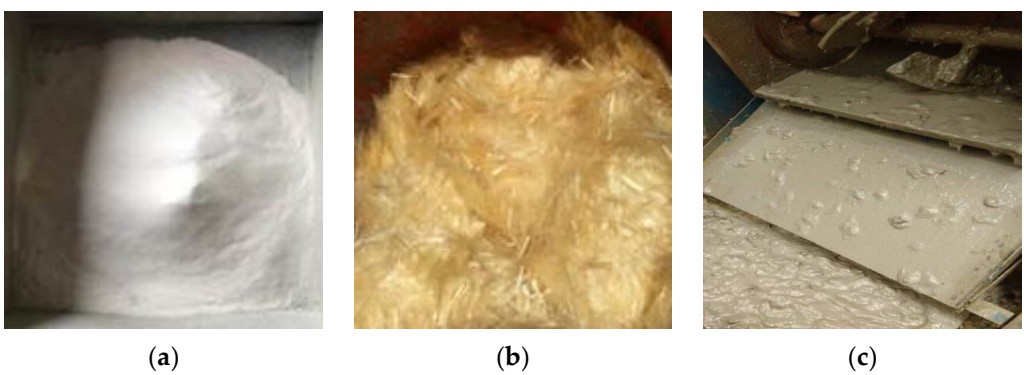

| **(a)** | **(b)** | **(c)** |

**Figure 1.** Test materials: (**a**) Quartz sand, (**b**) PVA fiber, (**c**) Mixing of ECC with PVA fibers.

Table 3 shows the ratio of the C30 and C50 strength levels of regular concrete used in the test. Ordinary concrete is prepared using conventional silicate cement P.O 42.5, domestic water, fineness modulus of 2.3–2.6 ordinary sand, particle size of 5–25 mm continuous graded natural gravel, and a Poly carboxylic acid system of high-efficiency water reduction agent.

**Table 3.** Ordinary concrete strength mix (unit: kg/m$^3$).

| Grade of Concrete Strength | Cement | Water | Sand | Stone | Water Reducer | Water Cement Ratio |
|---|---|---|---|---|---|---|
| C30 | 400.0 | 212.0 | 800.0 | 1200.0 | 0.00 | 0.53 |
| C50 | 520.0 | 182.5 | 706.0 | 1177.0 | 1.41 | 0.35 |

### 2.2. Molding Process

This study used two phases of casting to produce functional gradient concrete examples., illustrated in Figure 2, with dimensions of 150 mm × 150 mm × 150 mm. The thicknesses of the ECC material are 75 mm and 45 mm, respectively, while the thicknesses of regular concrete are 75 mm and 105 mm. Table 4 shows the dosage of each material component of the ECC. Table 5 shows the composition of each material component of concrete.

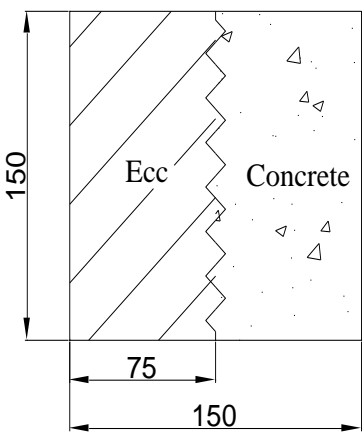 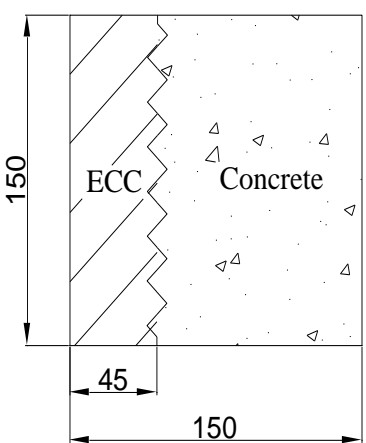

**Figure 2.** Test piece fabrication (unit: mm).

**Table 4.** ECC material components (unit: kg/m$^3$).

| ECC Thickness (mm) | Cement | Fly Ash | Silica Fume | Quartz Sand | Water | Water Reducer | Fiber |
|---|---|---|---|---|---|---|---|
| 75 | 1337 | 4010 | 1337 | 2407 | 2339.8 | 26.7 | 144.8 |
| 45 | 801.9 | 2405.7 | 801.9 | 1443.42 | 1403.3 | 16.04 | 86.8 |

**Table 5.** Material components of ordinary concrete (unit: kg/m$^3$).

| Strength of Concrete | Cement | Water | Sand | Stone | Water Reducer |
|---|---|---|---|---|---|
| C30 (75 mm) | 223 | 118 | 446 | 668 | 0 |
| C30 (105 mm) | 312 | 165 | 624 | 936 | 0 |
| C50 (75 mm) | 290 | 102 | 394 | 656 | 4.8 |
| C50 (105 mm) | 405 | 142 | 550 | 917 | 7.5 |

An HJS-60 double horizontal axis concrete test mixer is used to create specimens for this test. The steps for casting specimens are as follows:

(1) First, ensure that the mixer is clean on the inside, and then moisten the mixer bin wall with water to prevent it from becoming too dry and absorbing the water in the ECC mix.

(2) Weigh the amount of fly ash, silica fume, quartz sand, cement into the mixer, uniform dry mixing 60 s; then pour into the cement, mixing 30 s; then into the water to the mixer, mixing 3 min; and then add Poly carboxylic acid system of high-efficiency water reducing agent, mixing 2 min; and finally, the PVA fiber is poured into the mixer, stirring for 2 min, so that the PVA–ECC has good mobility and bonding properties.

(3) After mixing, use a steel ruler to complete part of the ECC thickness (75 mm and 45 mm) pouring size in the lower part of the plastic mold, and then place the plastic mold on the vibration table for vibration compactness. Vibration should continue until the ECC evenly spreads the corresponding size of the test mold.

(4) Clean the residual PVA fiber in the mixer with water, pour the weighed amount of each material of ordinary concrete into the mixer successively, and mix evenly and dry for 60 s; then put the water into the mixer and mix for 3 min, and then

complete the whole concrete casting in the test model, at this point, place the test film on the vibrating table and hold it down with your hand to vibrate for about 2 min, scraping the surface flat. At the same time, place a plastic film on the surface of the specimen to prevent surface moisture from evaporating until the entire specimen casting is complete.

(5)   Specimens were numbered, de-molded after 24 h of natural curing at room temperature, relocated to a conventional curing environment (temperature $20 \pm 2$ °C, relative humidity 95%) for 28 days, and tested after the stipulated duration.

### 2.3. Interface Processing Methods

The quality of interfacial bonding has a direct influence on the transmission of stress between the reinforcing body and the matrix. It significantly impacts the overall mechanical characteristics of the composite materials. Insufficient interfacial bonding in a composite material may lead to interfacial debonding damage when subjected to stress, hence limiting the reinforcing action of the fiber. By correctly modifying the surface of the reinforcing material, it is possible to enhance not only the interlaminar shear strength of the composite material but also its tensile strength and modulus. Therefore, this research study utilizes suitable interface improvement technology to strengthen the bonding zone between the functional layers of functional gradient concrete and boost its overall mechanical qualities.

Three different interface treatments are used: the first is without special treatment, denoted by JM0 for comparison; the second is groove treatment, denoted by JM1. The third is fine steel wire mesh treatment, denoted by JM2, with both interfaces used to achieve a more reliable bond between the two materials.

It is noteworthy to mention that the interfacial enhancement technology of "fine wire mesh" is employed to improve the macroscopic and mechanical properties of functional gradient concrete by enhancing the interfacial bonding zone between the functional layers and "treating the interface between ECC and ordinary concrete layers". These characteristics are mechanical.

### 2.4. Loading Program

Due to the installation of functional gradient concrete, smaller skew and eccentricity will unavoidably occur. While its geometric center and physical center do not match, these factors are different degrees of its measured results. Therefore, this study avoids using an axial tensile test to determine the tensile strength of functional gradient concrete, instead using the range of 300 kN of the hydraulic universal materials testing machine to test the specimen splitting tensile strength following the provisions of the "Standard for Test Methods of Mechanical Properties of Ordinary Concrete" (GB/T 50081-2019) [36]. The appearance of the specimen was evaluated for major faults before the test, and the exact location of the splitting interface was calculated based on the various diameters of the two materials. The curved pads were put at the top and bottom of the splitting bond surface of the functional gradient concrete, centered at the top and bottom, respectively, in the splitting tensile strength test. Figure 3 depicts the specimen loading design as well as the real arrangement.

The specific procedure for the splitting tensile strength test of functionally graded concrete is as follows:

(1)   After removing the specimens from the curing location, clean the surface of the specimen with the upper and lower bearing plate surfaces.

(2)   The specimen will be placed in the center of the test machine under the pressure plate; the split pressure surface and split surface should be perpendicular to the top surface of the specimen molding and marked with a marker to mark split surface; the specimen is cushioned with circular arc-shaped pads and cushion strips in the upper and lower pressure plates, and the cushions and cushion strips should be with the specimen on the center line below the center line aligned with the top of the top surface of molding and vertical.

(3) Start the testing machine, and when the upper-pressure plate and the arc-shaped pad come together, adjust the ball seat to ensure a balanced contact. The loading should be even and continuous, with a loading speed of 0.05–0.08 MPa/s. The adjustment of the test machine throttle should be stopped until the specimen is close to destruction, and then the destructive load value, accurate to 0.1 MPa, should be recorded. The hydraulic universal material splitting test machine is depicted in Figure 4.

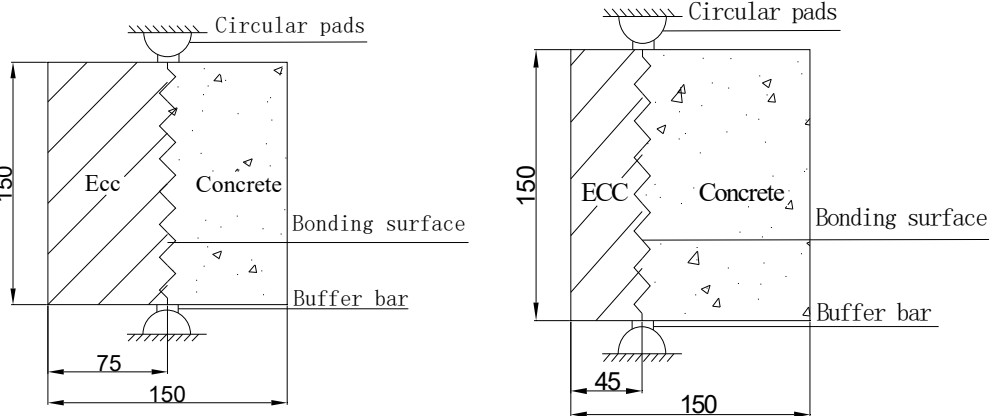

**Figure 3.** Specimen loading diagram (unit: mm).

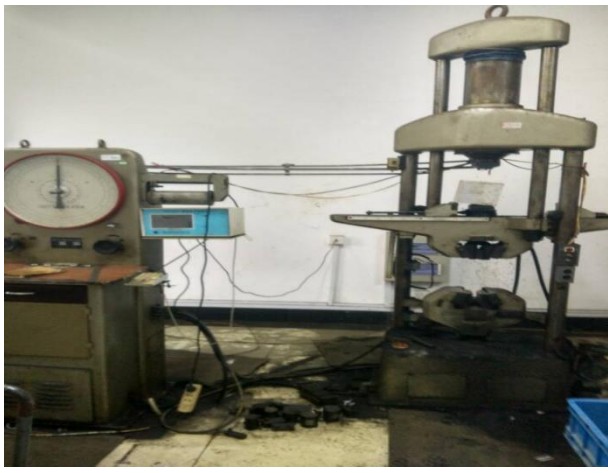

**Figure 4.** Hydraulic universal material splitting tester.

The following equation can calculate functional gradient concrete splitting tensile strength:

$$f = \frac{2F}{\pi A} = 0.637\frac{F}{A} \tag{1}$$

where $f$ is the splitting tensile strength (MPa); $F$ is the specimen failure load (kN); $A$ is the area of the splitting surface of the specimen (mm$^2$); $A = 150$ mm $\times$ 150 mm.

## 3. Test Damage Pattern Phenomena and Results

### 3.1. Test Phenomena

Figure 5 illustrates the occurrence of splitting tensile testing of functional gradient concrete. The interface of the splitting damage of the functional gradient concrete specimen was largely detected at the interfacial bond surface of the two, and the damaged surface between the two materials was reasonably straight in the test. As illustrated in Figure 5, the load has just started to load onto the specimen splitting surface at the beginning of the initial period in the splitting strength test. At this time, it is nearly impossible to see the appearance of evident fissures with the naked eye, and the specimen is in normal operation.

The specimen progressively began to appear on the split surface on the left and right sides of the short and thin little fractures as the weight increased. With increasing load, the length and width of the cracks at the splitting surface increased and became the first main crack.

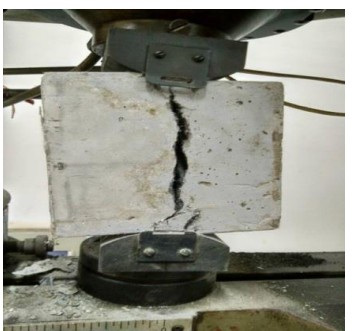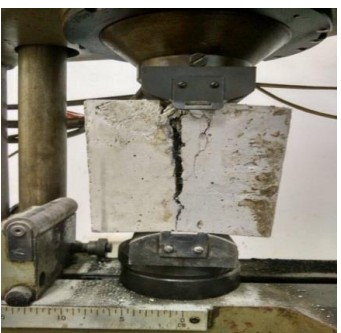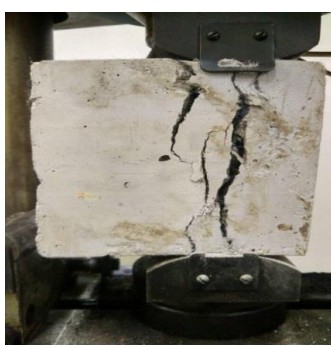

**Figure 5.** Specimen test phenomena.

In contrast, small cracks appeared around the main crack and gradually extended downward, and there were no obvious cracks in other parts of the specimen far away from the splitting surface. Finally, when the specimen was put near the destructive load, the splitting surface divided it into two parts: concrete on one side and ECC on the other, accompanied by a loud "bang" sound. Figures 5 and 6 demonstrate that some crushed concrete is dislodged near the loading end, exposing some aggregate and paste or crushing some ECC material near the loading end. This phenomenon may be caused by pure tensile forces in the middle and compressive stresses in the specimen's upper and lower loading areas; furthermore, the dislodging phenomenon did not happen because of the PVA fibers' limiting tensile tensions.

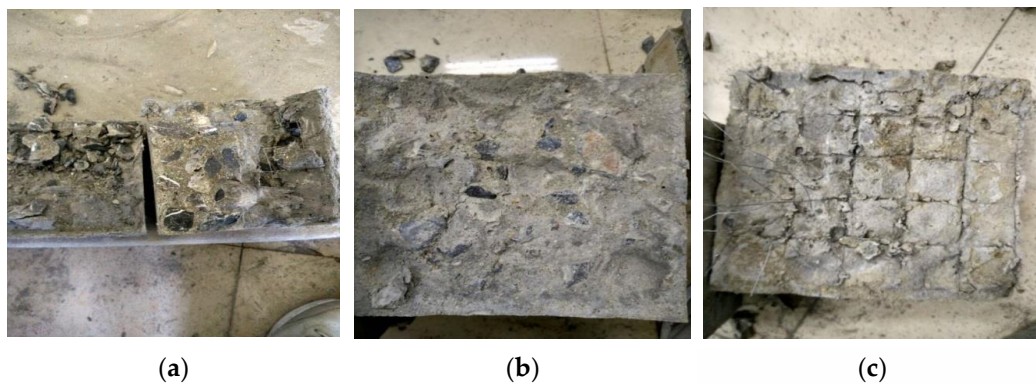

(**a**)           (**b**)           (**c**)

**Figure 6.** Specimen damage pattern, (**a**) Concrete crushing, (**b**) JM1 interface type damage pattern, (**c**) JM2 interface type damage pattern.

The damage morphology of the functional gradient concrete contact is depicted in Figure 6. Upon further observation, the test revealed that when an external load is applied to ordinary concrete, the functional gradient concrete interface splits, causing more scattered particles to fall off and some particles in the compressed state to a crisp. In contrast, the ECC material's ECC surface appears to have short and thin independent cracks due to the presence of PVA fibers rather than large particles scattered. An interface that has not undergone special treatment, with a relatively straight specimen damage surface, relies solely on the penetration and adhesion of the two materials; an interface that uses groove processing enhancement technology will inevitably find some concrete embedded in the ECC, suggesting that the two materials should achieve a better connection; steel wire mesh spacing also appeared to exhibit varying degrees of fracturing and deformation, indicating that the fine steel wire mesh for the specimen to assume a certain degree of loading. The

interface using steel wire mesh treatment of enhancement technology has also been cleaved into the break, resulting in the steel wire mesh spacing extrusion.

*3.2. Test Results*

The splitting tensile strength of the functional gradient concrete was determined using Equation (1) based on the destructive load recorded during the test. The average value was determined to be the splitting tensile strength of the group of specimens; the results are displayed in Table 6 below.

**Table 6.** Splitting tensile strength test results of functionally graded concrete.

| Specimen Number | Splitting Tensile Strength | | |
| --- | --- | --- | --- |
| | Failure Load (kN) | Strength (MPa) | Average Value (MPa) |
| C30-JM0-75 mm | 50.2<br>47.1<br>48.6 | 1.421<br>1.333<br>1.376 | 1.376 |
| C30-JM1-75 mm | 71.3<br>68.4<br>67.2 | 2.019<br>1.936<br>1.903 | 1.953 |
| C30-JM2-75 mm | 51.7<br>48.1<br>54.4 | 1.464<br>1.362<br>1.540 | 1.455 |
| C30-JM0-45 mm | 40.5<br>44.1<br>35.2 | 1.145<br>1.249<br>0.997 | 1.130 |
| C30-JM1-45 mm | 56.0<br>52.0<br>48.6 | 1.585<br>1.472<br>1.376 | 1.478 |
| C30-JM2-45 mm | 47.4<br>46.2<br>46.7 | 1.342<br>1.308<br>1.322 | 1.324 |
| C50-JM0-75 mm | 60.5<br>63.0<br>62.1 | 1.713<br>1.784<br>1.758 | 1.752 |
| C50-JM1-75 mm | 73.0<br>81<br>72.4 | 2.067<br>2.293<br>2.050 | 2.14 |
| C50-JM2-75 mm | 66.2<br>62.7<br>59.8 | 1.874<br>1.775<br>1.693 | 1.781 |
| C50-JM0-45 mm | 103.8<br>106.4<br>86.4 | 2.939<br>3.012<br>2.446 | 2.799 |
| C50-JM1-45 mm | 117.0<br>108.5<br>112.0 | 3.312<br>3.072<br>3.171 | 3.185 |
| C50-JM2-45 mm | 107.0<br>94.1<br>100.3 | 3.029<br>2.664<br>2.840 | 2.844 |

Note: C30 and C50 indicate the strength grade of concrete; JM0 indicates no special treatment; JM1 indicates artificially created groove treatment; and JM2 indicates fine steel wire mesh treatment. 75 mm and 45 mm denote the thickness of the ECC material.

## 4. Analysis of Test Results

This section focuses on the splitting strength test of functional gradient concrete. It examines the impact of three parameters—concrete strength grade, interfacial reinforcement technology, and changes in ECC thickness–on its splitting performance and the corresponding trends of influence.

### 4.1. Effect of Concrete Strength Class

The functional gradient concrete splitting test data in Table 6 is used to assess the trend and degree of influence of various concrete strength grades on the specimens' splitting tensile strength. It was discovered that the interfacial bond strength between layers of functional gradient concrete rises with an increase in concrete strength grade, irrespective of changes made to the interfacial reinforcement procedure or the ECC thickness.

A correlation between the functionally graded concrete's splitting tensile strength and the concrete strength grade is illustrated in Figure 7. This correlation illustrates how the splitting tensile strength of the specimens is influenced by the various concrete strength grades, as shown in Table 6. As will be elaborated in the following section, it has been discovered that the splitting tensile strength of the specimens is influenced differently by the interfacial reinforcement processes JM0 (interface without special treatment), JM1 (interface with notch treatment), and JM2 (interface with wire mesh treatment).

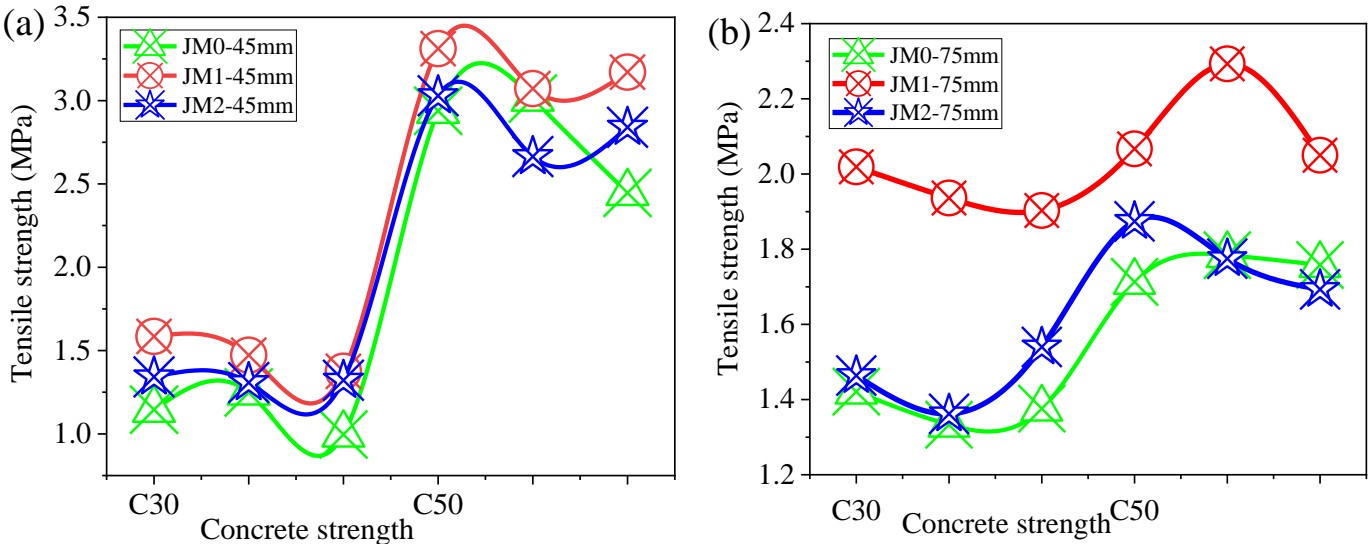

**Figure 7.** Effect of different concrete strength classes on splitting tensile strength. (**a**) ECC 45 mm case, and (**b**) ECC 75 mm case.

Indeed, this study conducted tests on the performance of interfacial bond strength in functional gradient concrete interlayer. These tests focused on different interfacial reinforcement processes, namely JM0, JM1, and JM2. Due to space limitations, only the test results are provided below to aid in the analysis of Figure 7. The average interlayer interfacial bond strength of the specimen corresponding to C30 is 2.965 MPa, while the specimen corresponding to C50 has an average interlayer interfacial bond strength of 4.186 MPa. The specimen JM0-45 mm is used as an illustration. The specimen corresponding to C30 has an average interlayer interfacial bond strength of 3.522 MPa, while the specimen corresponding to C50 has an average interlayer interfacial bond strength of 4.459 MPa.

Additionally, there is a 0.937 MPa increase in the interlayer interfacial bond strength between the C30 and C50 specimens. Figure 7 demonstrates that various treatments applied at the boundary between concrete and ECC have a substantial impact on the splitting tensile strength of functionally graded concrete. Specifically, the use of JM1 has a more noticeable effect compared with the use of JM2.

The aforementioned experimental analysis demonstrates that the functional gradient concrete interlayer interfacial bond strength increases with conventional concrete strength grade under the same interfacial reinforcement procedure and ECC thickness. The concrete strength grade parameter significantly influences the interfacial bond strength between layers of functional gradient concrete, which could be primarily due to the fact that in part of the interfacial zone between conventional concrete strength and ECC, higher strength grades of concrete contribute more to the interfacial bond strength of functional gradient concrete.

Table 6 shows that when the concrete strength class is increased from 30 to C50, the splitting tensile strengths corresponding to JM0-45 mm are 1.130 MPa and 2.799 MPa; those corresponding to JM1-45 mm are 1.478 MPa and 3.185 MPa; those corresponding to JM2-45 mm are 1.324 MPa and 2.844 MPa; those corresponding to JM0-75 mm are 1.376 MPa and 1.752 MPa; those corresponding to JM1-75 mm are 1.953 MPa and 2.14 MPa; the corresponding splitting tensile strength of JM2-75 mm are 1.455 MPa and 1.781 MPa, which makes it easy to detect that the splitting tensile strength of the specimens has increased to varied degrees. Still, the magnitude of the increase has stayed consistent. The specimens' increases in split tensile strength were, in that order, 147.7%, 115.5%, 114.8%, 27.3%, 9.58%, and 22.4%. When the interface is treated as a notch, and the thickness of the ECC is 45 mm, the increase in the split tensile strength of the specimen is 1.707 MPa; when the interface is treated as a notch, and the thickness of the ECC is 75 mm, the increase in the split tensile strength of the specimen is 0.187 MPa. The primary cause of this might be attributed to the weight of the concrete in the functional gradient concrete specimens as well as the fact that the splitting tensile properties of the concrete are more strongly affected by changes in its strength grade. Consequently, it is evident that an increase in concrete strength grade significantly impacts the splitting tensile strength of functional gradient concrete.

### 4.2. Effect of the Interfacial Reinforcement Process

The quality of the interfacial bonding has a direct impact on the stress transfer effect between the matrix and the reinforcing body, which also has a bigger effect on the macroscopic mechanical properties of the composite materials. If the interfacial bonding is too weak, the composite material is subject to interfacial debonding damage under stress, and the fiber cannot properly express the reinforcing action. By appropriately changing the surface of the reinforcing material, the composite material's interlaminar shear strength, tensile strength, and modulus can all be increased. As a result, appropriate interfacial reinforcement technology is employed in this study to reinforce the interfacial bonding zone between functional layers of functional gradient concrete in order to improve its macroscopic mechanical properties.

As seen in Figure 8, the test in this research study uses three different approaches to treat the interface between ECC and regular concrete layers. The specimens underwent a maximum of 30 min of treatment. The 150 mm × 150 mm interlayer surface served as the bonding surface, and in order to meet the strength requirements at the interface bond, the concrete surface and the ECC surface had separate treatments.JM0 denotes that the interface is not given any special treatment; JM1 denotes that artificial grooves measuring 25 mm in diameter and 10 mm in depth are formed at the interface of the first material poured; and JM2 denotes the installation of a fine steel wire mesh with a 20 mm grid spacing at the interface of the first goods poured.

Figure 9 shows the scanning electron microscopy (SEM) images of the interfacial bonding zone under various concrete strength grades, interfacial treatments, and ECC thicknesses, displaying the microscopic morphology of the hydration products at the interfacial bonding zone of the samples. According to Figure 8, after 28 days, both the concrete and the ECC reinforcement, acting as the matrix and reinforcement, had sufficiently hydrated the interface, resulting in the appearance of many Ca(OH)2 crystals, hydrated calcium silicate (C-S-H) gel, and calcium sulfoaluminate in the interfacial bonding zone and

the absence of an obvious interfacial transition zone, as shown in Figure 10. The hydration products have also achieved the "embedded solid" state and are well-bonded; altogether, the bonding effect between the two is good, particularly in the case of JM0, where the JM1 treatment is particularly evident.

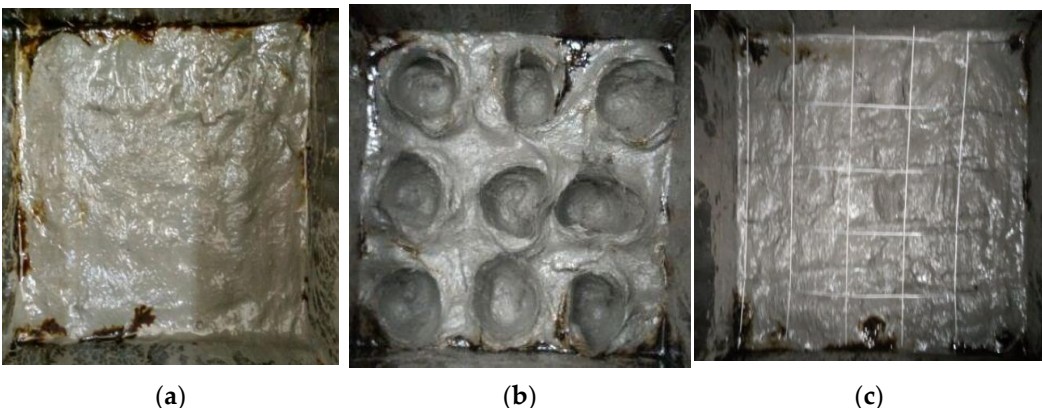

**Figure 8.** Interface between ECC and plain concrete layer (**a**) JM0–Without special treatment, (**b**) JM1–Artificial groove treatment, (**c**) JM2-Fine wire mesh treatment.

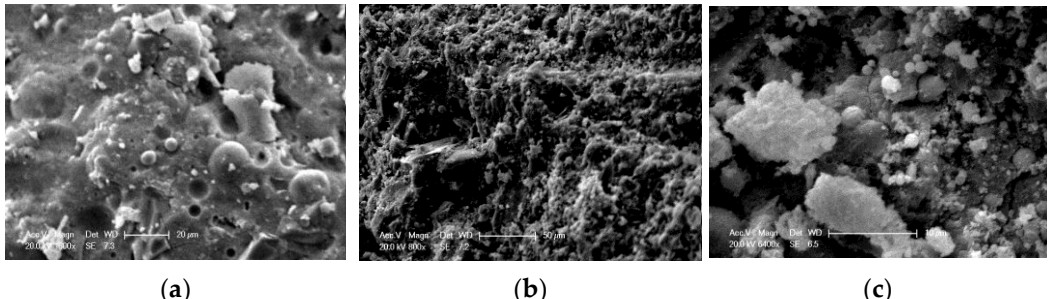

**Figure 9.** Microscopy of hydration products in the functional layer's interfacial bonding zone for C30 concrete strength (**a**) C30-JM0-45 mm, (**b**) C30-JM1-45 mm, (**c**) C30-JM2-45 mm.

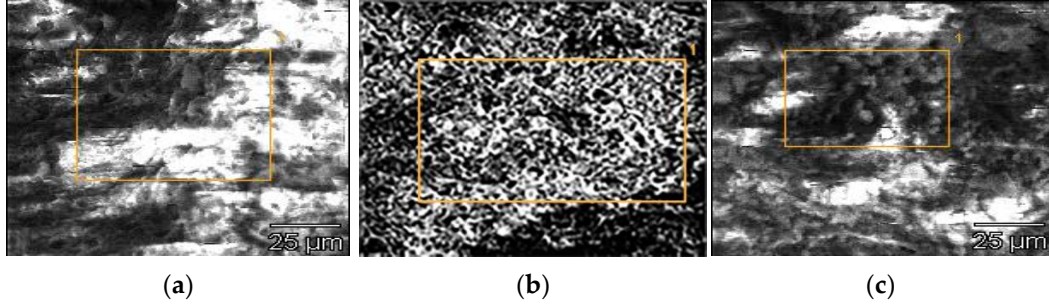

**Figure 10.** Hydration product marking corresponding to Figure 9. (**a**) Hydration product area at ECC, (**b**) Hydration product area at interface bond, (**c**) Hydration product area at concrete.

Referring back to Figure 8, in the instance of JM2 treatment, the steel wire mesh treatment at the functional layer's interface causes, to some extent, the emergence of fine, short cracks that are visible to the unaided eye at the interface of the ECC and the concrete between the layers and across their surfaces, thereby disrupting the continuous and uniform distribution of the hydration products. Compared with C30-JM2-30 mm, the length and width of the cracks in C30-JM2-45 mm are more noticeable in smaller regions. This suggests that the interface treatment and ECC thickness have little influence on the microscopic composition of hydration products at the interface but have a considerable effect on the microscopic morphology.



The test demonstrated that different interfacial treatment effects resulted in different values of splitting tensile strength of functionally graded concrete when comparing the average splitting tensile strength of specimens with the same parameters, such as concrete strength grade and ECC thickness, but with different interfacial enhancement processes. When the interface enhancement process is JM0 (no special treatment), JM1 (grooving treatment), and JM2 (fine steel wire mesh treatment), the corresponding four groups of specimens splitting tensile strength values are 1.13 MPa, 1.478 MPa, 1.324 MPa; 1.376 MPa, 1.953 MPa, 1.455 MPa; 2.799 MPa, 3.185 MPa, 2.844 MPa; 1.752 MPa, 2.14 MPa, and 1.781 MPa. When the comparison in Figure 6 is taken into consideration, it is discovered that the specimen cleavage tensile strength of the chosen interfaces increases by only 0.029–0.194 MPa when fine steel wire mesh treatment (JM2) is applied instead of the unspecialized treatment (JM0), which is not a statistically significant increase, is not readily apparent. This might be because, despite the fine wire mesh being applied to the specimen's interface, the fine wire mesh and the load on the specimen were essentially kept parallel, meaning that the fine wire mesh's function was ineffectively fulfilled. The specimen's split tensile strength increased noticeably, reaching 0.348–0.577 MPa, after the interface was changed from the unspecialized treatment (JM0) to the grooving treatment (JM1). The cause is primarily due to the fact that the concrete and ECC to achieve a better "embedded solid", concrete and ECC contact area increases, and consequently, the interface between the two mechanical occlusion forces is greater, reflected in the macro-expression is the function of gradient concrete splitting tensile strength of the ensuing increase. It is discovered that increasing the specimen's interface roughness will improve its splitting tensile strength regardless of how the interface process is improved. Moreover, there is an obvious correlation between the interface enhancement process and the splitting performance of functional gradient concrete. As such, careful consideration must be given to the way the specimen's interface is handled.

### 4.3. Effect of ECC Thickness

This section investigates the effect of ECC thickness on the specimens' splitting tensile characteristics. When the concrete strength grade and interfacial reinforcement process parameters remain constant with the increase in ECC thickness from 45 mm to 75 mm, the corresponding specimen splitting tensile strengths are 1.13 MPa, 1.376 MPa; 1.478 MPa, 1.953 MPa; 1.324 MPa, 1.455 MPa in order, and the specimen splitting tensile strength increased by 0.246 MPa, 0.478 MPa, 0.131 MPa; 2.799 MPa, 1.752 MPa; 3.185 MPa, 2.14 MPa; 2.844 MPa, 1.781 MPa, respectively, and the reductions of specimens was 37.4%, 32.81%, and 37.38%, respectively. As can be shown in Figure 11a, the change in ECC thickness has a significant impact on the splitting tensile strength of functional gradient concrete.

The test findings indicate that the variation in ECC thickness is a significant determinant of the splitting tensile strength of functional gradient concrete. However, the splitting and tensile characteristics of the specimens do not exhibit a rise as the thickness of ECC increases. In fact, they show contrasting patterns. Figure 11a demonstrates that the splitting tensile strength of the specimens exhibited a rising pattern when the concrete strength grade of C30 was used. Conversely, Figure 11b illustrates a sequential drop in the splitting tensile strength of the specimens when the concrete strength grade of C50 was utilized.

The phenomenon may be attributed to the simultaneous increase in the concrete strength grade and the thickness of ECC. As the concrete strength increases, the bonding between the cement paste, sand, and gravel in the concrete mix weakens. This lack of cohesion in the mix reduces the fluidity, resulting in poor bonding between the concrete and ECC at the interface region. Consequently, there may be some weakening at the interface, leading to a decrease in the splitting and tensile properties of the specimens.

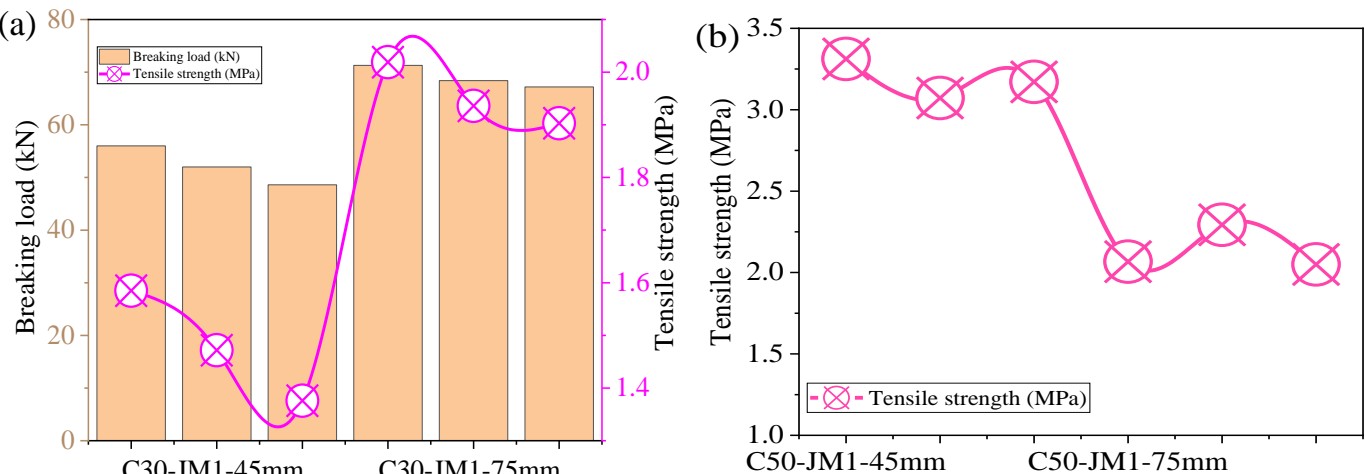

**Figure 11.** Effect of different ECC thicknesses on splitting tensile strength. (**a**) C30 case, and (**b**) C50 case.

### 4.4. Elaboration/Comparison with Past Literature

ECC stands out among gelling materials due to its exceptional ability to withstand high levels of tensile strain. Prior research on the tensile qualities of the material has been conducted by Kanda et al. [37], Li [38], Lee et al. [39], and other researchers. In a recent research study conducted by Chen et al. [40], the impact of various dosages (ranging from 0.25 to 1.0 vol%) of recycled tire polymer fiber (RTPF) on the splitting tensile characteristics of ECC was examined. The study also studied the influence of strain rates on these properties. RTPF was used as a substitute for polyvinyl alcohol fiber (PVAF) in the ECC samples. In their study, Qasim et al. [41] examined how different dosages (ranging from 0.25 to 1.0 vol%) of recycled tire polymer fiber (RTPF) affected the splitting tensile properties of ECC. They replaced a portion of the PVAF -ECC and steel-polyvinyl alcohol hybrid fiber-reinforced ECC (SPH-ECC) with RTPF. The researchers conducted experiments to investigate the interfacial bond strength and compared the effects of hybrid fiber ECC with single fiber ECC. The goal of their study was to identify promising and effective retrofit materials for reinforced concrete structures. Ouyang et al. [42] and Gao et al. [43] examined the impact of surface ECC residual bond area damage on the split tensile strength of the repair system. The findings indicated a decline in the interfacial binding strength as the temperature increased. In their study, Tawfek et al. [44] examined how the orientation of fibers impacts the mechanical characteristics of ECC composites. They used two distinct casting techniques for their investigation.

Refs. [40–44] are derived from split tests, in which literature [40,41] examined the impact of various fiber dopings on the split tensile characteristics of ECC. These dopings are directly relevant to the dopings discussed in this study. The impact of ECC interfacial bond strength on the split tensile characteristics of ECC was examined in previous studies [42,43], which is directly related to Section 4.2 of the current study. This study described in the literature [44] used SEM and digital image correlation (DIC) to examine the damage process in ECC specimens subjected to compression and tensile testing, which is relevant to Section 4.3 of the current article. Ultimately, the research conducted in this publication may serve as a valuable point of reference for future investigations, particularly when comparing them to well-established studies.

It should be noted that when studying materials with multiple variables, such as specimen design, engineering background, and influence parameters, the comparative analysis can only be conducted through research methodology and design theory. The research theory serves as a mere indication.

## 5. Conclusions

This study examines the extent of the impact of three variables, namely, the grade of concrete strength, interfacial reinforcing technique, and variation in ECC thickness, on the tensile characteristics of functionally graded concrete. It also analyzes the damage patterns effect of ECC concrete via splitting tests. Subsequently, the main findings are as follows:

(1) The splitting tensile strength of functional gradient concrete rises as the concrete strength grade increases.

(2) Varied approaches to the interface between concrete and ECC have a substantial impact on the splitting tensile strength of functional gradient concrete, with the influence of JM1 being more pronounced compared with the use of JM2.

(3) The variation in ECC thickness is a crucial determinant of the splitting tensile strength of functional gradient concrete. Nevertheless, the splitting tensile characteristics of the specimens do not exhibit a correlation with the thickness of ECC, demonstrating distinct patterns.

**Author Contributions:** Conceptualization, X.Z.; Formal analysis, X.X.; Investigation, P.L.; Writing—original draft, Y.X.; Writing—review & editing, Q.L. All authors have read and agreed to the published version of the manuscript.

**Funding:** This study was funded by the Wuhan City College Teaching and Research Project (grant number 2023CYYBJY013) and Scientific Research Project (grant number 2022CYYBKY02), the National Natural Science Foundation of China (grant numbers 52178182, 52108262, and U1934217), Science and Technology Research and Development Program Project of China Railway Group Limited (Major special project, No.: 2020-Special-02, 2021-Special-08, 2022-Special-09; Major project, No.: 2021-Special-02; Key project, No.: 2021-Key-11, No.: 2022-Key-46). The authors also have received research grants from the Natural Science Foundation for Distinguished Young Scholars of Hunan Province (2022JJ10075), the Natural Science Foundation of Hunan Province of China (2020JJ5982), and the Hunan Science and Technology Plan Project (2023SK2014).

**Institutional Review Board Statement:** Not applicable.

**Informed Consent Statement:** Not applicable.

**Data Availability Statement:** All relevant data are within the study.

**Acknowledgments:** The authors thank the Wuhan City College Teaching and Research Project (grant numbers 2023CYYBJY013) and Scientific Research Project, the National Natural Science Foundation of China, the Science and Technology Research and Development Program Project of China Railway Group Limited, the National Science Foundation for Distinguished Young Scholars of Hunan Province, the Natural Science Foundation of Hunan Province of China and the Hunan Innovative Province Construction Special Project for funding this research, as well as anonymous reviewers for their contribution to this study.

**Conflicts of Interest:** Authors Liu Qiang, Zhang Xiaoqiang, and Xu Xiaofeng was employed by the company Wuhan United Investment Real Estate Co., Ltd., Jinan Municipal Engineering Design & Research Institute (Group) Co., Ltd., and Hubei United Investment Group. The remaining authors declare that the research was conducted in the absence of any commercial or financial relationships that could be construed as a potential conflict of interest.

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
