# Peer review of "Splitting Tensile Test of ECC Functional Gradient Concrete with PVA Fiber Admixture"

_coatings, doi:10.3390/coatings14020231_

Round 1
Reviewer 1 Report
Comments and Suggestions for Authors
1. “Currently, engineered cementitious composites (ECC) replace traditional ordinary concrete and are applied in engineering structures, one of the main ways to solve the above problems” Which above problems?
2. “The use of ECC itself has a nonlinear deformation capacity, energy absorption capacity, and crack control ability, coupled with the interface between ECC and concrete after the interface treatment technology, has a good bonding work performance, so that it becomes a kind of functional composites [3], the common bearing engineering structure, improve the member's Load carrying capacity and ductility [4], the crack width of the member has also been better controlled [5], thus improving the service life of the member, but also for the seismic design of engineering structures provides a new structural form.” This sentence is too long and meaningless.
3. “Given this, in order to obtain the axial tensile strength of ECC, the split tensile strength test was considered to indirectly reflect the axial tensile strength of ECC [7].” This statement is common for all concrete and not only for ECC.
4. “The study of strength's basic mechanical properties shows that the manifold mainly affects its splitting tensile strength.” What is strength’s basic mechanical properties? Strength is a part of mechanical properties.
5. “Ibrahim et al. [15] pointed out that the main influencing factors of the bonding properties of new and old concrete are the interface treatment method and state, the type of interfacial agent, the bond strength of reinforced concrete, and the difference in deformation of new and old concrete.” What is new and old concrete? Why suddenly this kind of concrete came?
6. The authors mentioned the particle size of quartz sand as 100 mm to 200 mm; it will be better to provide particle distribution curves for the different materials used for a better understanding.
7. What do the authors mean by class 1 fly ash? Kindly provide the chemical compositions of the fly ash, silica fume and other cementitious materials used in the study.
8. The total cementitious material in the study is about 1200kg/m3; what about the strength achieved by this proportion?
9. What do the authors mean by the second horizontal casting employed for the study? Authors may consider to rewrite the methodology used properly.
10. Kindly check the mix proportion provided in Table 4.
11. What is the need to test the specimen loaded as per Fig 2 a since the split plane doesn’t pass through the interface of two mixes?
12. Fig 4 b and c do not show a proper split plane; then how can the authors confirm the specimen failed due to split tensile stress?
13. There is no sign of fiber bridging the failed specimen, kindly include the proper photographs with proper fiber bridging.
14. Mark the circular arc-shaped pads and cushion strips in the diagram.
15. In line 303, the authors mentioned the JM0-30mm specimen; kindly check this.
16. How do the authors directly correlate the split tensile stress to interfacial bond strength?
17. The comparative values provided in line 30The and comparative values provided in lines 304-305 and 310-311 are confusing; kindly check this.
18. What is the design strength of functional gradient concrete?
19. For Fig 6, the tensile strength graph alone is sufficient; there is no need to provide the load values.
20. In the test specimens, fiber is only provided for functional gradient concrete; how does this fiber enhance the overall split strength of the specimen?
21. The three different approaches used to treat the interface between ECC and regular concrete layers should be written under the methodology part before table 6. Since from table 6 it is impossible to identify what is JM0, Jm1 and JM2 means.
22. Is the SEM image taken for the ECC layer or the ordinary concrete layer? Kindly mark the mentioned hydrated products in the SEM micrographs.
23. “This might be because, despite the fine wire mesh being applied to the specimen's interface, the fine wire mesh and the load on the specimen were essentially kept 398 parallel, meaning that the fine wire mesh's function was ineffectively fulfilled’ this sentence is confusing. Does the author mean that there is no significance in providing mesh?
24. What does author mean by “two mechanical bite forces,” in line 404.
25. From Table 6, the authors mentioned that the average stress value for C50-JM1-45mm was 3.185Mpa, but this is not shown in the provided graph. Also, the authors should give a valid justification for the reduction in strength when the thickness of the ECC layer increased to 75 mm.
26. The results provided in Table 6 and Fig 9 do not match; kindly check this.
27. Please add an elaboration/comparison with past literature for the discussion part.
28. The conclusion part is too long. Please consider shortening the sentences and just give the essential results.
Comments on the Quality of English Language
Most the sentences are too long and difficult to understand.
Author Response
Reviewer 1
Response to Reviewers' Comments (Manuscript ID: coatings-2842022)
Title: Splitting tensile test of ECC functional gradient concrete with PVA fiber admixture
Author's Response
Dear Editors and Reviewers,
Thank you for allowing us to submit a revised version of the manuscript "Splitting tensile test of ECC functional gradient concrete with PVA fiber admixture (Manuscript ID: coatings-2842022)" for publication in Coatings. We sincerely appreciate the time and effort you and the reviewers dedicated to providing feedback on our manuscript and are grateful for our paper's insightful and constructive comments. The thorough review helped immensely in shaping and improvement of the manuscript. We have incorporated most of the suggestions made by the reviewers. Please see below; Fonts in blue have been provided for a point-by-point response to the reviewers' comments and concerns. Also, it should be noted that all of the editors' and reviewers' suggestions have been applied to the paper in addition to the reviewer's comments.
- “Currently, engineered cementitious composites (ECC) replace traditional ordinary concrete and are applied in engineering structures, one of the main ways to solve the above problems” Which above problems?
Author’s Response: Thank you for your constructive comment. In the opening paragraph of the report, the author includes relevant information and emphasizes it by using the color red. The following are the additions:
Concrete materials are commonly utilized in civil engineering, roads, bridges, and other engineering domains. However, it is important to acknowledge the presence of defects in concrete that can hinder its application. For instance, the tensile strength of steel reinforcement combined with concrete is significantly lower (only 1/8-1/12 of the compressive strength), leading to early corrosion of the longitudinal reinforcement. Additionally, the inherent brittleness of concrete further exacerbates these defects, impeding the progress of concrete construction materials.
- “The use of ECC itself has a nonlinear deformation capacity, energy absorption capacity, and crack control ability, coupled with the interface between ECC and concrete after the interface treatment technology, has a good bonding work performance, so that it becomes a kind of functional composites [3], the common bearing engineering structure, improve the member's Load carrying capacity and ductility [4], the crack width of the member has also been better controlled [5], thus improving the service life of the member, but also for the seismic design of engineering structures provides a new structural form.” This sentence is too long and meaningless.
Author’s Response: Thank you for pointing these out. In the opening paragraph of the report, the author includes relevant information and emphasizes it by using the color red. The following are the additions:
By harnessing the nonlinear deformation, energy absorption, and crack control properties of the ECC and optimizing the interface between ECC and concrete to enhance its bond performance, it transforms into a functional composite material [3]. This results in enhanced load-carrying capacity and ductility of the structural element, as well as improved control over crack width, thereby extending the service life of the element [4-5].
- “Given this, in order to obtain the axial tensile strength of ECC, the split tensile strength test was considered to indirectly reflect the axial tensile strength of ECC [7].” This statement is common for all concrete and not only for ECC.
Authors’ Response: We thank the reviewer for bringing this to our attention. The author acknowledges your issue and has included the pertinent information below, shown in red:
Furthermore, researchers have conducted a range of empirical and theoretical investigations on the tensile characteristics and ontological connections of ECC [7-9], however a cohesive consensus has not yet been established. Thus, this research examines the splitting tensile strength test for functional gradient concrete as a means to assess its axial tensile strength indirectly. The test results produced from this splitting tensile strength test exhibit low variability, and the procedure is straightforward and easily achievable.
[7]Yu, J., Lu, C., Chen, Y., & Leung, C. K. (2018). Experimental determination of crack-bridging constitutive relations of hybrid-fiber Strain-Hardening Cementitious Composites using digital image processing. Construction and Building Materials, 173, 359-367.
[8]Li, Y., Guan, X., Zhang, C., & Liu, T. (2020). Development of high-strength and high-ductility ECC with saturated multiple cracking based on the flaw effect of coarse river sand. Journal of Materials in Civil Engineering, 32(11), 04020317.
[9]Yao, Q., Teng, X., Lu, C., Sun, H., Mo, J., & Chen, Z. (2023). Influence of accelerated chloride corrosion on mechanical properties of sea sand ECC and damage evaluation method based on nondestructive testing. Journal of Building Engineering, 63, 105520.
- “The study of strength's basic mechanical properties shows that the manifold mainly affects its splitting tensile strength.” What is strength’s basic mechanical properties? Strength is a part of mechanical properties.
Authors’ Response: We thank the reviewer for this important comment. The author's phrasing is unclear and has been revised as follows and emphasized in red.
Research on the splitting tensile strength of functionally graded concrete has shown that several elements influence its splitting tensile strength.
- “Ibrahim et al. [15] pointed out that the main influencing factors of the bonding properties of new and old concrete are the interface treatment method and state, the type of interfacial agent, the bond strength of reinforced concrete, and the difference in deformation of new and old concrete.” What is new and old concrete? Why suddenly this kind of concrete came?
Authors’ Response: We thank the reviewer for these questions. The majority of the damage in functional gradient concrete mostly occurs at the interfacial bond. Consequently, the interface plays a crucial role in determining the splitting and tensile characteristics of functional gradient concrete. In this context, new concrete is specifically defined as "construction and demolition waste generated during new construction activities. "whereas old concrete refers to " solid harmful or non-harmful waste generated from the construction, renovation & restoration and, development & demolition of small or large building structures, bridges, piers, roads, dams, etc."
- The authors mentioned the particle size of quartz sand as 100 mm to 200 mm; it will be better to provide particle distribution curves for the different materials used for a better understanding.
Authors’ Response: We thank the reviewer for bringing this to our attention.
|
(a) Silica fume |
(b) PVA fiber |
(c) Mixing of ECC with PVA fibers |
Figure 1. Test materials.
- What do the authors mean by class 1 fly ash? Kindly provide the chemical compositions of the fly ash, silica fume and other cementitious materials used in the study.
Authors’ Response: Thank you for pointing these out.:The author acknowledges your issue and has included the pertinent information below, shown in red:
Fly ash is composed of SiO2, Al2O3, Fe2O3, and CaO. For further details, please refer to the website http://www.glhuayue.com/index.php?aid=469. The chemical makeup of silica fume consists mostly of amorphous SiO2, which makes up 92.4% of its composition. For further details, go to the website http://www.aerbadi-nxyl.com/custom/a/33.
- The total cementitious material in the study is about 1200kg/m3; what about the strength achieved by this proportion?
Authors’ Response: The reviewer made a great point. The ECC test used in this research is representative of two specific strength categories of concrete, namely C30 and C50, with the corresponding mixing ratios provided in Table 3.
- What do the authors mean by the second horizontal casting employed for the study? Authors may consider to rewrite the methodology used properly.
Authors’ Response: The authors would like to apologize for any confusion. The casting process should consist of two phases. The original text should be modified to read: This work used two phases of casting to produce functional gradient concrete examples.
- Kindly check the mix proportion provided in Table 4.
Authors’ Response: We thank the reviewer for the question. Table 4 has been checked.
- What is the need to test the specimen loaded as per Fig 2 a since the split plane doesn’t pass through the interface of two mixes?
Authors’ Response: Thank you for pointing these out. Concerning your questions, the authors make every effort to respond. As the split face passes through a cross-section of both mixtures, the splitting tensile test of ECC concrete transforms into the tensile test of conventional concrete. Determining the weakest form of stress in ECC concrete is the objective of the current testing procedure.
Actually, this study comprises four experiments: "A performance evaluation of interlayer interfacial bonding in functional gradient concrete," "An experimental investigation into the splitting and tensile properties of functional gradient concrete," "An experimental study into the elastic modulus properties of functional gradient concrete," and "An interfacial characterization and theory of interfacial phenomena in functional gradient concrete."
The test specimens utilized in this investigation are visualized in Figure 2.
- Fig 4 b and c do not show a proper split plane; then how can the authors confirm the specimen failed due to split tensile stress?
Authors’ Response: Thank you for pointing these out. The author is following the split tensile test protocols, has redrawn Figure 4 below, and has provided the test photos of split planes. Thank you.
- There is no sign of fiber bridging the failed specimen; kindly include the proper photographs with proper fiber bridging.
Authors’ Response: You have raised an important point here. The authors have redrawn Figures 4 and 5, and the experimental phenomena concerning fiber bridging are given in the micrographs of Figure 9.
|
(a) Concrete crushing |
(b) JM1 interface type damage pattern |
(c) JM2 interface type damage pattern
|
- Mark the circular arc-shaped pads and cushion strips in the diagram.
Authors’ Response: Thank you for pointing these out. The author redraws Figure 2 below. Thank you.
Figure 2. Specimen loading diagram.
- In line 303, the authors mentioned the JM0-30mm specimen; kindly check this.
Authors’ Response: Thank you for pointing these out. The author rewrote this paragraph and section 4.1. Thank you.
- How do the authors directly correlate the split tensile stress to interfacial bond strength?
Authors’ Response: We appreciate the reviewer for this insightful and smart comment.
Actually, this study comprises four experiments: "A performance evaluation of interlayer interfacial bonding in functional gradient concrete," "An experimental investigation into the splitting and tensile properties of functional gradient concrete," "An experimental study into the elastic modulus properties of functional gradient concrete," and "An interfacial characterization and theory of interfacial phenomena in functional gradient concrete."
Furthermore, the correlation between the interface bond strength and split tensile stress is evident. This is primarily due to the increased contact area between the concrete and ECC, which facilitates a more secure "embedded" connection between the two materials. As a result, the interface between the two mechanical occlusion forces becomes more substantial, as evidenced by the macro-expression of the functional gradient of the concrete split tensile strength that follows this increase.
- The comparative values provided in line 30The and comparative values provided in lines 304-305 and 310-311 are confusing; kindly check this.
Authors’ Response: Thank you for pointing these out. This paragraph and section 4.1 were rewritten by the author. Thank you.
- What is the design strength of functional gradient concrete?
Authors’ Response: Thank you for pointing these out. The ECC test used in this research is representative of two specific strength categories of concrete, namely C30 and C50, with the corresponding mixing ratios provided in Table 3.
- For Fig 6, the tensile strength graph alone is sufficient; there is no need to provide the load values.
Authors’ Response: Thank you for pointing these out. 修改图6如下As follows, alter Figure 6:
Figure 6. Effect of different concrete strength classes on splitting tensile strength. (a) Case of ECC 45mm case, and (b) Case of ECC 75mm
- In the test specimens, fiber is only provided for functional gradient concrete; how does this fiber enhance the overall split strength of the specimen?
Authors’ Response: Thank you for pointing these out. The factors contributing to the total tensile strength of fiber-reinforced ECC concrete are as follows:
(1) The inclusion of polyvinyl alcohol (PVA) fiber in cement is beneficial owing to its exceptional elasticity, strength, modulus, and compatibility with cement. The fiber acts as a bridge inside the material.
(2) When ECC concrete experiences cracking, it undergoes sustained expansion rather than rapid destruction. This expansion leads to the formation of many tightly packed micro-cracks, which effectively control the width of the cracks. Unlike regular concrete, the crack width in ECC can be regulated and is not unpredictable.
- The three different approaches used to treat the interface between ECC and regular concrete layers should be written under the methodology part before table 6. Since from table 6 it is impossible to identify what is JM0, Jm1 and JM2 means.
Authors’ Response: Thank you for pointing these out. The factors contributing to the total tensile strength of fiber-reinforced ECC concrete are as follows:
(1) The inclusion of polyvinyl alcohol (PVA) fiber in cement is beneficial owing to its exceptional elasticity, strength, modulus, and compatibility with cement. The fiber acts as a bridge inside the material.
(2) When ECC concrete experiences cracking, it undergoes sustained expansion rather than rapid destruction. This expansion leads to the formation of many tightly packed micro-cracks, which effectively control the width of the cracks. Unlike regular concrete, the crack width in ECC can be regulated and is not unpredictable.
- Is the SEM image taken for the ECC layer or the ordinary concrete layer? Kindly mark the mentioned hydrated products in the SEM micrographs.
Authors’ Response: Thank you for pointing these out. The SEM image presented in Figure 8 illustrates the hydration products' microscopic morphology within the interfacial binding zone of the sample. Additionally, Figure 9 contains SEM micrographs that have been annotated with references to the hydrate.
|
(a) Hydration product area at ECC |
(a) Hydration product area at interface bond |
(a) Hydration product area at concrete |
Figure 9. Hydration product marking corresponding to Figure 8.
- “This might be because, despite the fine wire mesh being applied to the specimen's interface, the fine wire mesh and the load on the specimen were essentially kept 398 parallel, meaning that the fine wire mesh's function was ineffectively fulfilled’ this sentence is confusing. Does the author mean that there is no significance in providing mesh?
Authors’ Response: Thank you for pointing these out.
Initially, the loads exerted on both the wire mesh and the specimen maintain a consistent parallel alignment, as dictated by the loading technique.
Furthermore, the primary purpose of fine wire mesh is to improve the interface between ECC (Engineered Cementitious Composite) and regular concrete layers. This is achieved by utilizing the interface enhancement technology of fine wire mesh, which increases the bonding area between the different layers of functional gradient concrete. As a result, the overall macroscopic and mechanical properties of the concrete are improved. Properties related to the behavior and characteristics of mechanical systems.
The quality of the interfacial bonding has a direct impact on the transmission of stress between the reinforcing body and the matrix. It also significantly influences the overall mechanical characteristics of the composite material. If the interfacial bond is insufficiently strong, the composite material is susceptible to interfacial debonding damage when subjected to stress, and the fiber is unable to fully use its reinforcing effect. By correctly modifying the surface of the reinforcing material, not only can the interlaminar shear strength of the composite material be enhanced, but also its tensile strength and modulus may be increased.
- What does author mean by “two mechanical bite forces,” in line 404.
Authors' Response: The authors would like to apologize for any confusion. The sentence is rewritten as follows and red highlighted in the paper: This is primarily due to the fact that the concrete and ECC to achieve a better "embedded solid", concrete and ECC contact area increases, and consequently the interface between the two mechanical occlusion force is greater, reflected in the macro-expression is the function of gradient concrete splitting tensile strength of the ensuing increase.
- From Table 6, the authors mentioned that the average stress value for C50-JM1-45mm was 3.185Mpa, but this is not shown in the provided graph. Also, the authors should give a valid justification for the reduction in strength when the thickness of the ECC layer increased to 75 mm.
Authors’ Response: Thank you for pointing these out. The test findings indicate that the variation in ECC thickness is a significant determinant of the splitting tensile strength of functional gradient concrete. However, the splitting and tensile characteristics of the specimens do not exhibit a rise as the thickness of ECC increases. In fact, they show contrasting patterns. Figure 11(a) demonstrates that the splitting tensile strength of the specimens exhibited a rising pattern when the concrete strength grade of C30 was used. Conversely, Figure 11(b) illustrates a sequential drop in the splitting tensile strength of the specimens when the concrete strength grade of C50 was utilized.
The phenomenon may be attributed to the simultaneous increase in the concrete strength grade and the thickness of ECC. As the concrete strength increases, the bonding between the cement paste, sand, and gravel in the concrete mix weakens. This lack of cohesion in the mix reduces the fluidity, resulting in poor bonding between the concrete and ECC at the interface region. Consequently, there may be some weakening at the interface, leading to a decrease in the splitting and tensile properties of the specimens.
- The results provided in Table 6 and Fig 9 do not match; kindly check this.
Authors’ Response: Thank you for pointing these out. The author reproduces figure 9 below. Thank you.
- Please add an elaboration/comparison with past literature for the discussion part.
Authors’ Response: Thank you for pointing these out. An additional section 4.4 has been appended in an endeavor by the authors to provide an answer to your question.
ECC stands out among gelling materials due to its exceptional ability to withstand high levels of tensile strain. Prior research on the tensile qualities of the material has been conducted by Kanda et al. [38], Li [39], Lee et al. [40], and other researchers. In a recent research conducted by Chen et al. [41], the impact of various dosages (ranging from 0.25 to 1.0 vol%) of recycled tire polymer fiber (RTPF) on the splitting tensile characteristics of Engineered Cementitious Composites (ECC) was examined. The study also studied the influence of strain rates on these properties. RTPF was used as a substitute for polyvinyl alcohol fiber (PVAF) in the ECC samples. In their study, Qasim et al. [42] examined how different dosages (ranging from 0.25 to 1.0 vol%) of recycled tire polymer fiber (RTPF) affected the splitting tensile properties of Engineered Cementitious Composites (ECC). They replaced a portion of the PVAF -ECC and steel-polyvinyl alcohol hybrid fiber-reinforced ECC (SPH-ECC) with RTPF. The researchers conducted experiments to investigate the interfacial bond strength and compared the effects of hybrid fiber ECC with single fiber ECC. The goal of their study was to identify promising and effective retrofit materials for reinforced concrete structures. Ouyang et al. [43] and Gao et al. [44] examined the impact of surface ECC residual bond area damage on the split tensile strength of the repair system. The findings indicated a decline in the interfacial binding strength as the temperature increased. In their study, Tawfek et al. [45] examined how the orientation of fibers impacts the mechanical characteristics of ECC composites. They used two distinct casting techniques for their investigation.
The Ref. [41-45] are derived from split tests, in which literature [41, 42] examined the impact of various fiber dopings on the split tensile characteristics of ECC. These dopings are directly relevant to the dopings discussed in this study. The impact of ECC interfacial bond strength on the split tensile characteristics of ECC was examined in previous studies [43, 44], which is directly related to section 4.2 of the current work. The work described in literature [45] used scanning electron microscopy (SEM) and digital image correlation (DIC) to examine the damage process in ECC specimens subjected to compression and tensile testing, which is relevant to section 4.3 of the current article. Ultimately, the research conducted in this publication may serve as a valuable point of reference for future investigations, particularly when comparing them to well-established studies.
It should be noted that when studying materials with multiple variables, such as specimen design, engineering background, and influence parameters, the comparative analysis can only be conducted through research methodology and design theory. The research theory serves as a mere indication.
[38]Kanda, T., Lin, Z., & Li, V. C. (2000). Tensile stress-strain modeling of pseudostrain hardening cementitious composites. Journal of Materials in Civil Engineering, 12(2), 147-156.
[39]Lee, B. Y., Kim, J. K., & Kim, Y. Y. (2010). Prediction of ECC tensile stress-strain curves based on modified fiber bridging relations considering fiber distribution characteristics. Computers & Concrete, 7(5), 455-468.
[40]Li, V. C. (2003). On engineered cementitious composites (ECC) a review of the material and its applications. Journal of advanced concrete technology, 1(3), 215-230.
[41]Chen, M., Jiang, R., Zhang, T., Zhong, H., & Zhang, M. (2024). Development of green engineered cementitious composites with acceptable dynamic split resistance utilising recycled polymer fibres. Construction and Building Materials, 415, 134979.
[42]Qasim, M., Lee, C. K., & Zhang, Y. X. (2022). An experimental study on interfacial bond strength between hybrid engineered cementitious composite and concrete. Construction and Building Materials, 356, 129299.
[43]Ouyang, J., Guo, R., Wang, X. Y., Fu, C., Wan, F., & Pan, T. (2023). Effects of interface agent and cooling methods on the interfacial bonding performance of engineered cementitious composites (ECC) and existing concrete exposed to high temperature. Construction and Building Materials, 376, 131054.
[44]Gao, S., Zhao, X., Qiao, J., Guo, Y., & Hu, G. (2019). Study on the bonding properties of Engineered Cementitious Composites (ECC) and existing concrete exposed to high temperature. Construction and Building Materials, 196, 330-344.
[45]Tawfek, A. M., Ge, Z., Yuan, H., Zhang, N., Zhang, H., Ling, Y., ... & Šavija, B. (2023). Influence of fiber orientation on the mechanical responses of engineering cementitious composite (ECC) under various loading conditions. Journal of Building Engineering, 63, 105518.
[46]Yang, G., Dong, Z., Bi, J., Zhao, K., & Li, F. (2023). Experimental study on the dynamic splitting tensile properties of polyvinyl-alcohol-fiber-reinforced cementitious composites. Construction and Building Materials, 383, 131233.
- The conclusion part is too long. Please consider shortening the sentences and just give the essential results.
Authors’ Response: Thank you for pointing these out. The conclusion was revised by the author. Thank you.
This study examines the extent of impact of three variables, namely, the grade of concrete strength, interfacial reinforcing technique, and variation in ECC thickness, on the tensile characteristics of functionally graded concrete. It also analyzes the pattern of effect via splitting tests. Subsequently, the further deductions were made:
(1) The splitting tensile strength of functional gradient concrete rises as the concrete strength grade increases.
(2) Varied approaches to the interface between concrete and ECC have a substantial impact on the splitting tensile strength of functional gradient concrete, with the influence of JM1 being more pronounced compared to the use of JM2.
(3) The variation in ECC thickness is a crucial determinant of the splitting tensile strength of functional gradient concrete. Nevertheless, the splitting tensile characteristics of the specimens do not exhibit a correlation with the thickness of ECC, demonstrating distinct patterns.
The authors express their gratitude to the reviewing experts for their important suggestions to enhance the article.

Reviewer 2 Report
Comments and Suggestions for Authors
Dear Authors,
I read an interesting article on gradient reinforced cements. My overall assessment is positive. A very solid review of the literature on the subject has been carried out. I consider the research methodology to be appropriate. Regarding the preparation of the samples, it is a pity that only two ECC layer thicknesses were tested. What puzzles me, however, is what is new in this article. The components used are familiar, the cementitious gradient materials too. I think the authors should emphasise the importance of the research better.
My mertorical observation is that cement hydration can be well assessed by performing FTIR studies in addition to SEM studies. Perhaps the authors should consider supplementing the studies.
Reading through the manuscript, I also noticed a few editorial comments
line 150 - class I fly ash - lowercase
line 154 - fine steel wire - lowercase
line 158, 180 - should be "polycarboxylic"
line 248 - the load - lowercase
In addition, please standardise the captions of the figures. I believe that Figures 4 and 5 show the destruction of samples exactly as shown in Figure 2. I think that the above figures should be better captioned.
Best regards
Reviewer
Author Response
Reviewer 2
Response to Reviewers' Comments (Manuscript ID: coatings-2842022)
Title: Splitting tensile test of ECC functional gradient concrete with PVA fiber admixture
Author's Response
Dear Editors and Reviewers,
Thank you for allowing us to submit a revised version of the manuscript "Splitting tensile test of ECC functional gradient concrete with PVA fiber admixture (Manuscript ID: coatings-2842022)" for publication in Coatings. We sincerely appreciate the time and effort you and the reviewers dedicated to providing feedback on our manuscript and are grateful for our paper's insightful and constructive comments. The thorough review helped immensely in shaping and improvement of the manuscript. We have incorporated most of the suggestions made by the reviewers. Please see below; Fonts in blue have been provided for a point-by-point response to the reviewers' comments and concerns. Also, it should be noted that all of the editors' and reviewers' suggestions have been applied to the paper in addition to the reviewer's comments.
Dear Authors,
I read an interesting article on gradient reinforced cements. My overall assessment is positive. A very solid review of the literature on the subject has been carried out. I consider the research methodology to be appropriate. Regarding the preparation of the samples, it is a pity that only two ECC layer thicknesses were tested. What puzzles me, however, is what is new in this article. The components used are familiar, the cementitious gradient materials too. I think the authors should emphasise the importance of the research better.
Authors’ Response: The authors have made substantial revisions to the paper in response to your positive feedback. These changes aim to enhance the study's significance and deliver something of value to the readers. Further investigation will be conducted by the author in subsequent studies to delve deeper into the subject matter.
1# mertorical observation is that cement hydration can be well assessed by performing FTIR studies in addition to SEM studies. Perhaps the authors should consider supplementing the studies.
Authors’ Response: This is a valid point regarding cement hydration research, and we concur that FTIR analysis can be useful in this regard. However, time constraints dictate that the authors will address this issue in subsequent research. The authors express their gratitude to the reviewing experts for their important suggestions to enhance the article.
2# Reading through the manuscript, I also noticed a few editorial comments
line 150 - class I fly ash – lowercase
Authors’ Response: Thank you for pointing this out. The paper has already been revised. Thank you.
line 154 – fine steel wire – lowercase
Authors’ Response: Thank you for pointing this out. The paper has already been revised. Thank you.
line 158, 180 - should be "polycarboxylic"
Authors’ Response: Thank you for pointing this out. The paper has already been revised. Thank you.
line 248 - the load – lowercase
Authors’ Response: Thank you for pointing this out. The paper has already been revised. Thank you.
In addition, please standardise the captions of the figures. I believe that Figures 4 and 5 show the destruction of samples exactly as shown in Figure 2. I think that the above figures should be better captioned.
Authors’ Response: Thank you for your constructive comment. The authors revised Figures 4 and 5.
The authors express their gratitude to the reviewing experts for their important suggestions to enhance the article.

Round 2
Reviewer 1 Report
Comments and Suggestions for Authors
Authors have taken care the comments carefully.